# Persistent extrasynaptic hyperdopaminergia in the mouse hippocampus induces plasticity and recognition memory deficits reversed by the atypical antipsychotic sulpiride

Jill Rocchetti[1], Caroline Fasano[1], Gregory Dal-Bo[1], Elisa Guma[1], Salah El Mestikawy[1,2], Tak-Pan Wong[1], Gohar Fakhfouri[1], Bruno Giros[1,3]*

1 Department of Psychiatry, Douglas Hospital, Mc Gill University, Montreal, Québec, Canada, 2 Sorbonne Université, INSERM, CNRS, NPS – IBPS, Paris, France, 3 Université Paris-Cité, INCC UMR 8002, CNRS, Paris, France

* bruno.giros@mcgill.ca

## Abstract

Evidence suggests that subcortical hyperdopaminergia alters cognitive function in schizophrenia and antipsychotic drugs (APD) fail at rescuing cognitive deficits in patients. In a previous study, we showed that blocking D2 dopamine receptors (D2R), a core action of APD, led to profound reshaping of mesohippocampal fibers, deficits in synaptic transmission and impairments in learning and memory in the mouse hippocampus (HP). However, it is currently unknown how excessive dopamine affects HP-related cognitive functions, and how APD would impact HP functions in such a state. After verifying the presence of DAT-positive neuronal projections in the ventral (temporal), but not in the dorsal (septal), part of the HP, GBR12935, a blocker of dopamine transporter (DAT), was infused in the CA1 of adult C57Bl/6 mice to produce local hyperdopaminergia. Chronic GBR12935 infusion in temporal CA1 induced a mild learning impairment in the Morris Water Maze and abolished long-term recognition memory in novel-object (NORT) and object-place recognition tasks (OPRT). Deficits were accompanied by a significant decrease in DAT$^+$ mesohippocampal fibers. Intrahippocampal or systemic treatment with sulpiride during GBR infusions improved the NORT deficit but not that of OPRT. *In vitro* application of GBR on hippocampal slices abolished long-term depression (LTD) of fEPSP in temporal CA1. LTD was rescued by co-application with sulpiride. In conclusion, chronic DAT blockade in temporal CA1 profoundly altered mesohippocampal modulation of hippocampal functions. Contrary to previous observations in normodopaminergic mice, antagonising D2Rs was beneficial for cognitive functions in the context of hippocampal hyperdopaminergia.

## Introduction

An excessive activity of the subcortical dopamine system has long been hypothesized to underlie the physiopathology of psychotic, or positive, symptoms of schizophrenia [1, 2]. Less, however, is known about hyperdopaminergia implication in cognitive defects associated with the disease.

**Data Availability Statement:** All relevant data are within the paper and its Supporting information files.

**Funding:** This work was supported by the Graham Boeckh Foundation for Schizophrenia Research and the Canadian Institute for Health Research in the form of grants to BG [2013090G-312343-PT; 201803PJT-399980-PT]. BG receives partial support in the form of a salary as a Canadian Research Chair in the Neurobiology of Mental Disorders at McGill University. GD-B and GF were supported by a postdoc grant from the Fonds de recherche du Québec (FRQS). The funders had no role in study design, data collection and analysis, decision to publish, or preparation of the manuscript.

**Competing interests:** The authors declare no competing interests.

Roughly 80% of individuals with a schizophrenia diagnosis suffer from cognitive deficits, which include alterations in working memory, executive function, and episodic memory. Working memory [3] is thought to rely on prefrontal activity in humans and be strongly modulated by dopamine [4]. The hippocampus has also been shown to play a critical role in these cognitive functions. For example, it plays a central role in creating long-term declarative memory [5] and is activated in humans during encoding and retrieval of novel information [6]. Interestingly, individuals with schizophrenia who often suffer from memory impairments show abnormal hippocampal activation based on PET and fMRI studies [7, 8]. Additionally, they also have volume, shape and subcellular abnormalities in the hippocampus [9]. Most research on schizophrenia alterations in hippocampal function focused primarily on glutamatergic alterations [9]. Although altered dopamine signalling plays a central role in the pathology of the disease, its role in this structure lacks thorough investigation.

In rodents, long-term spatial memory and novelty detection rely on hippocampus function [10, 11] and are strongly modulated by mesohippocampal connections and hippocampal dopamine receptors [12–14]. Dopamine acts on two classes of G-protein coupled receptors, D1-like—$D_1$/D5—and D2-like -$D_2$/$D_3$/D4—that have distinct downstream signaling pathways [15]. Even before dopamine neuromodulatory function was firmly established and dopamine receptors identified, antipsychotic drugs (APD) were used in clinical psychiatry to tame psychotic manifestations of schizophrenia. Advances in neurochemical imaging revealed that all APD share the pharmacological aspect of antagonizing D2-like dopamine receptors (D2Rs) with varying affinities [16]. This is still considered core feature of APD therapeutic effect toward psychosis [17]. However, APDs cope poorly with the cognitive symptoms of schizophrenia [18] and atypical drugs do not seem to do better than typical ones in this domain [19, 20]. In healthy individuals, whole-brain analyses have established a relationship between hippocampal D2Rs availability and episodic memory [21]. Previous work from our group revealed a negative outcome of altering D2Rs activity over the mesohippocampal pathway; blocking or removing D2Rs resulted in impairments in long-term potentiation (LTP) and depression (LTD) in CA1, as well as learning and memory deficits. Intriguingly, we also observed a prominent sprouting of dopamine transporter (DAT)-positive fibers (DAT$^+$) across all sublayers of temporal CA1. The deficits mainly relied on a lack of presynaptic D2 autoreceptor activity [22]. A separate group also confirmed that D2Rs are indispensable for hippocampal-dependent memory and plasticity at the CA3-CA1 synapse [23]. However, dopamine responses follow an inverted U-shape mode of action; too much or too little dopamine has proven detrimental while adequate functioning of dopaminoceptive areas require a well-balanced amount of dopamine receptor activity [4, 24]. If curtailing of D2Rs activity damages the homeostatic hippocampal functioning, would the latter still be the case if the hippocampus encountered pre-existing excess in dopamine activity?

Several models provide evidence that excessive D2Rs activation is indeed implicated in pathophysiological processes related to hyperdopaminergia in the hippocampus. In the endogenous hyperdopaminergic model of DAT knockout (DATKO) rodents [25], the hippocampus volume is significantly decreased [26, 27], LTP and LTD are impaired and there are deficits in cognitive flexibility in the spatial and cued Morris water maze. Interestingly, the typical antipsychotic haloperidol, a D2R-like antagonist, restores normal cognitive phenotype [28]. Rodent models of systemic exposure to psychostimulants, particularly cocaine and amphetamines, also provide evidence that dopamine reuptake inhibition impairs synaptic plasticity and cognitive processes in the hippocampus through a D2R-dependent signaling [29–31]. However, the regional effects of altered dopamine levels need to be further elucidated, since the hippocampus is engaged in polysynaptic wiring with other structures innervated by dopaminergic neurons [11, 32].

Given the lack of region-specific models of post-developmental dopamine excess, we performed intrahippocampal infusions of the DAT blocker 1-(2-Diphenylmethoxyethyl)-4-(3-phenylpropyl)piperazine (GBR12935) [33] in adult C57BL/6J mice. While we did not perform microdialysis in our study, GBR 12935 has long been established as the most selective DAT inhibitor amongst the GBR family with a high affinity for DAT (Kd~10nM), although very closely related to GBR12909 and GBR12921 in structure [34, 35]. In vivo microdialysis in the striatum or the nucleus accumbens of GBR-12935 as well as systemic or local administration of GBR-12909 show a rapid (within 20 minutes) and long-lasting increase of extracellular DA [36–40]. Chronic infusions in either the ventral or dorsal hippocampus of mice allowed studying the effects of local hyperdopaminergia on hippocampal-related memory functions and mesohippocampal connectivity.

In the present study, we did not aim or claim to mimic clinical conditions. Given the subcortical hyperdopaminergia theory of schizophrenia [1, 2], our goal was to investigate how localized hyperdopaminergia in the mouse hippocampus would affect memory-associated behavior in mice. In addition, in light of our previous findings on the detrimental role of D2 blockade on mesohippocampal synaptic transmission and on hippocampus-associated learning and memory under normodopaminergic condition [22], we studied if an antipsychotic, with D2 antagonism as its putative mechanism of action, could restore some of the cognitive impairments observed with hippocampal hyperdopaminergia.

Contrary to its deleterious effects in normodopaminergic animals, selective blockade of D2Rs in the hippocampus with the antipsychotic sulpiride turned out a beneficial intervention toward certain aspects of cognition and plasticity impaired by local hyperdopaminergia.

## Material and methods

### Animals

All experiments and animal care were performed according to the Canadian Council on Animal Care guidelines and approved by the Animal Care Committee of the Douglas Research Center (Animal Protocol # 2008–5570). All experiments were performed in accordance with ARRIVE guidelines (https://arriveguidelines.org). Two-month old males C57BL/6J mice were purchased from Charles Rivers. All mice were maintained in our local animal facility for at least a week before being used in experiments.

### Retrograde labeling

C57Bl/6J mice (2–4 months) were deeply anesthetized with isoflurane and placed in a stereotaxic apparatus (Kopf Instruments, Tujunga, CA, USA). Green and red retrobeads (200 nl; Lumafluor, Durham, NC) were unilaterally injected with a 0.5 μl syringe (Hamilton, Reno, USA) in the right and left hippocampi respectively at the following coordinates: AP: -3.2 mm, L: ±3.2 mm, DV: -3.3 mm for the ventral and (AP: -2 mm, L: ±1.5 mm, DV: -1.5 mm) for the dorsal sections. Four weeks after injections, animals were euthanized by decapitation (as per McGill SOP# 301 01-rodent euthanasia) after being placed into a C02 anesthetic chamber, and brain sections underwent immunolabeling.

### Immunohistolabeling and microscopy

For retrograde labeling analysis, animals were deeply anesthetized with ketamine injection (0.1 mg/g, i.p.) and perfused intracardially with 4% paraformaldehyde (PFA) in 0.1 M PBS (pH 7.4, 50 ml per mouse). Mouse brains were dissected, immersed in PFA overnight, and sliced the next day with a Vibratome into 30 μm coronal sections. All brain slices containing the ventral

mesencephalon were used for immunolabeling with DAT (rat monoclonal, 1/4000, Millipore, USA) and TH antibodies (rabbit, 1/4000, Millipore) as previously described [22]. Sections were rinsed, mounted on Superfrost+ slides and coverslipped with Fluoromount. Quantification of retrograde neurons was performed with an AxioObserver microscope (Zeiss, Germany) at 40x.

### Intrahippocampal infusions

Animals were stereotaxically cannulated with bilateral cannula (PlasticOne) to fit with the coordinates of dorsal (AP: -2; L: ± 1.5; DV: -1.5 mm from bregma) or ventral side (AP: -3.2; L: ±3; DV: -3) of CA1 region of the hippocampus. Animals recovered for 2 weeks prior to the start of infusions. Mice were infused with drugs daily (0.5µL/side/day) for 11 to 21 days depending on the experimental design and behavioral procedures. Infusion was performed in awake, freely moving animals 30 minutes prior to any behavioral testing.

### Drugs

$D_2/D_3$ antagonist (S)-(-)-Sulpiride, $D_1/D_5$ antagonist R-(+)-SCH23390 hydrochloride and 1(S),9(R)-(−)-Bicuculline methiodide were purchased from Sigma-Aldrich Canada. GBR12935 dihydrocloride was purchased from Tocris Bioscience. NaCl (.9%) was purchased commercially. For intracerebral infusions or systemic administration, solutions were adjusted at pH 7.2 and filtered before use each day. For electrophysiological experiments, drug stocks were kept at -80˚C, then diluted and bath-applied in artificial cerebrospinal fluid (aCSF) during recordings.

### Behavioral tests

Different groups of mice treated with either NaCl (.9%; 100mL/kg ip; 0.5µL/HP/day in intraHP infusions), sulpiride (50mg/kg ip; 15µmol/L, .5µL/HP/day in intraHP infusions), GBR12935 (300nM, .5µL/HP/day) or a combination of two treatments were submitted to various HP-dependent memory tests: spatial and cued versions of the Morris water maze (MWM), novel-object recognition task (NORT) and object-place recognition task (OPRT). See supplemental methods in S1 File for complete test description.

### Electrophysiological recordings

Field excitatory postsynaptic potentials (fEPSP) recordings were performed on CA3-CA1 glutamatergic synapses using artificial CSF as previously described [22] on coronal slices prepared from C57Bl/6J WT mice using artificial CSF in the presence of SCH23390 (1µmol/L), sulpiride (10µmol/L) and/or GBR12935 (30nmol/L). All experiments were done in the presence of $GABA_A$ receptor antagonist bicuculline (5µmol/L).

### Statistical comparisons

Statistical analyses were done using Student $t$ test, analysis of variance (ANOVA) or repeated-measure ANOVA (RMANOVA) with GraphPad (Prism, versions 6.02 and 8). ANOVAs were followed by post-hoc comparisons of the group means with Tukey HSD test for one-way ANOVA and Bonferroni-Dunn multiple-comparisons for RMANOVA. Significance criterion was set to α = .05. Results are reported as mean ± SEM (bars on figures denote SEM). All datasets passed D'Agostino & Pearson normality test.

Detailed procedures are found in Supplemental information.

## Results

### Differential DA innervation of the ventral and dorsal hippocampus

We previously reported a lack of DAT expression in the dorsal part of the hippocampus [22]. To confirm this specificity, and to see whether there were nevertheless DA projections in the dorsal hippocampus, we bi-laterally injected green and red fluorescent retrobeads into the CA1 of either the ventral or dorsal sections (S1 Fig in S1 File).

Four weeks later, we analyzed the retrogradely labeled neurons with antibodies against TH and DAT in the VTA and SNc. In the ventral hippocampus, we confirmed previous results showing that 81% of neurons projecting to the hippocampus are non-DA neurons, mostly originating from the VTA, 13% are DA neurons expressing TH and DAT, and 6% are DA neurons expressing TH but with no detectable DAT expression (Fig 1). Retrolabelling of neurons following dorsal injection of retrobeads show the same fraction (85%) of TH-negative neurons, and 15% of TH positive neurons all arising from the VTA, but not a single DAT positive neuron. This finding clearly confirms the absence of DAT immunoreactivity in the dorsal hippocampus, where there is nonetheless a significant DA innervation.

### GBR12935 infusion in the hippocampus induced impairment in habituation in the Morris water maze

We performed bilateral infusion of GBR12935 (300nmol/L) in CA1 of adult C57BL/6J mice in the ventral or the dorsal subregion of the hippocampus. As described above, DAT-positive neurons are only projecting to the ventral hippocampus, and we accordingly found GBR12935 to have no effect in dorsal hippocampus (S2 Fig in S1 File). Therefore, all following experiments were performed after ventral drug administration.

Separate groups of C57BL/6J mice were submitted to a daily intrahippocampal treatment with GBR12935 (300nmol/L; .5μL/HP) for 3 weeks. During the behavioral testing period, GBR12935 was infused daily 30 min prior to trials. Mice were submitted to 3 memory paradigms associated to hippocampal function: the Morris water maze (MWM), the novel-object recognition task (NORT) and the object-place recognition task (OPRT).

For the spatial version of the MWM, GBR infusions started at day 1 of learning phase. Mice were subjected to a learning procedure of 2 trials per day (Supplementary Methods in S1 File). GBR-treated mice exhibited a drastic decrease in escape latency within the first training days, as for sodium-chloride (NaCl) controls (Fig 2A). We extended the learning phase for 5 more days to confirm a slow-onset effect of chronic GBR on spatial learning. Two-way repeated-measure analysis of variance (RMANOVA) revealed a significant time x drug interaction in successful trials rate after 10 days of training ($F_{(9, 324)} = 1.995$, *p < .05). Comparison of the means confirmed that NaCl-treated mice displayed a significantly higher rate of successful trials than GBR-treated mice at day 9 (NaCl, .86±.05; GBR, .50±.09) (Fig 2B). Probe test of spatial memory performed 24h after the last training session revealed that both groups explored significantly more the target quadrant than the others.

Two-way ANOVA revealed a significant main effect of quadrant index over total variation ($F_{(3,144)} = 18.34$). Post hoc comparisons confirmed that both groups spent significantly more time in the platform quadrant than in the other quadrants (NaCl, 36.16%; GBR, 38.24%) (Fig 2C). No difference was detected between groups.

To test whether sensorimotor processing remained intact in GBR-treated animals, a small subset of naïve C57BL/6J mice infused with GBR for 7 days prior to first trial was submitted to a cued, or associative, version of the MWM. GBR-treated mice displayed equal learning

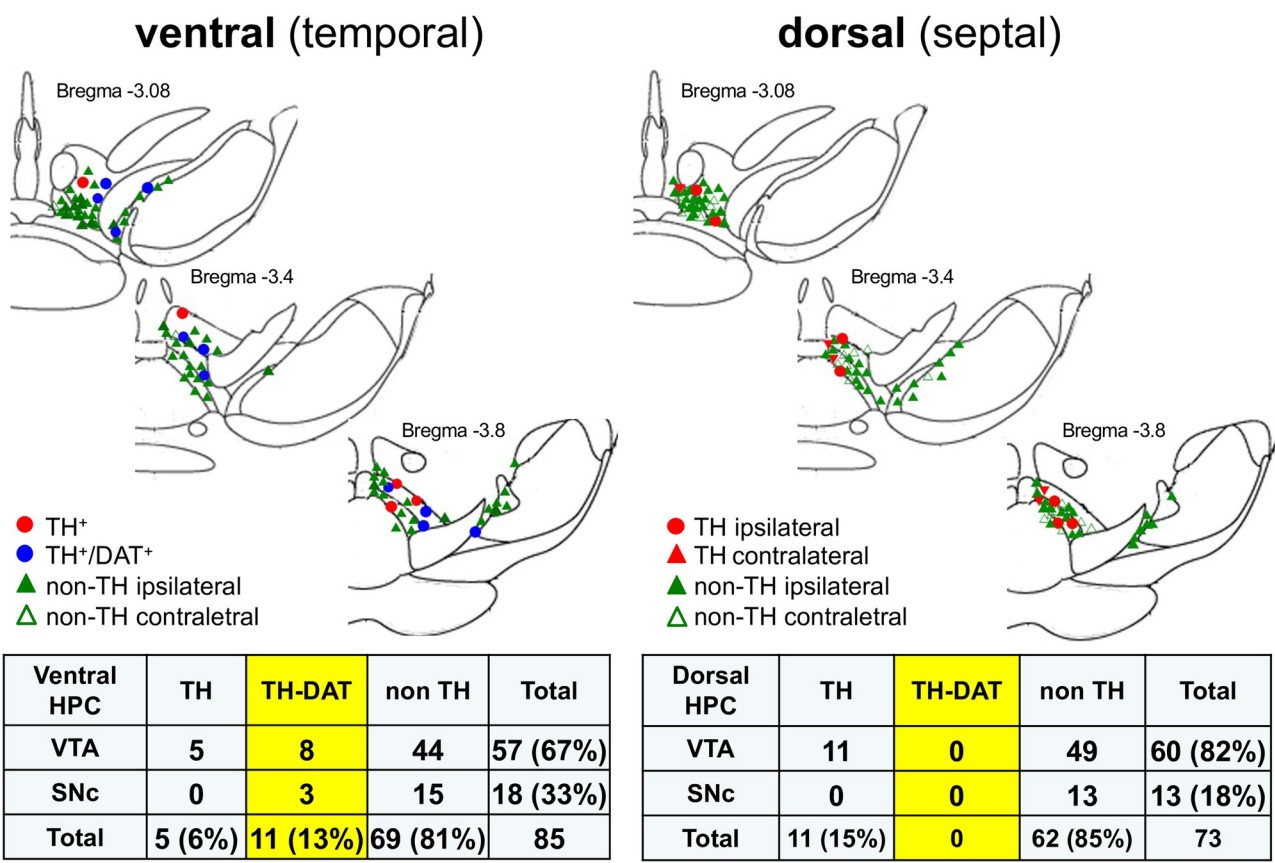

**Fig 1. Topographical organization of DA projections to the hippocampus.** Schematic representations of retrolabeled neuron distribution in ventral (left) and dorsal (right) mesencephalon of C57Bl/6J mice. Table compiling the numbers and proportion of TH and DAT-immunopositive neurons retrogradely labeled in ventral tegmental area (VTA) or substantia nigra pars compacta (SNc) following ventral (n = 10 mice) and dorsal (n = 10 mice) retrobeads bi-lateral administration.

performance compared with NaCl-treated mice after 5 days of training ((day5) NaCl, 15.25 sec; GBR, 19.29 sec) (Fig 2D).

## Chronic GBR12935 infusion abolished recognition memory and induced pruning of DA fibers

Chronically infused mice with GBR12935 in the ventral CA1 were submitted to spontaneous recognition tasks relying on the intact hippocampal function, including the NORT and OPRT. On day 1, animals were habituated to the squared open field and allowed to freely explore two identical objects for 5 minutes. On day 2, one object was replaced by a novel object with different shape and color (Fig 3A). Mice were subjected to the OPRT right after the NORT.

The same procedure was done at the beginning of the treatment period, and repeated after two weeks of daily GBR12935 infusion, with new pairs of objects for each test. A subset of animals treated for 3 weeks were submitted to a short-term version of the OPRT. One hour separated the presentation of the first configuration and the second one, where an object was moved (Fig 3A).

Time spent exploring the novel item (object or place) was significantly higher than the older one in NaCl and GBR-treated mice in both NORT (NaCl, .60±.01; GBR, .60±.01) and

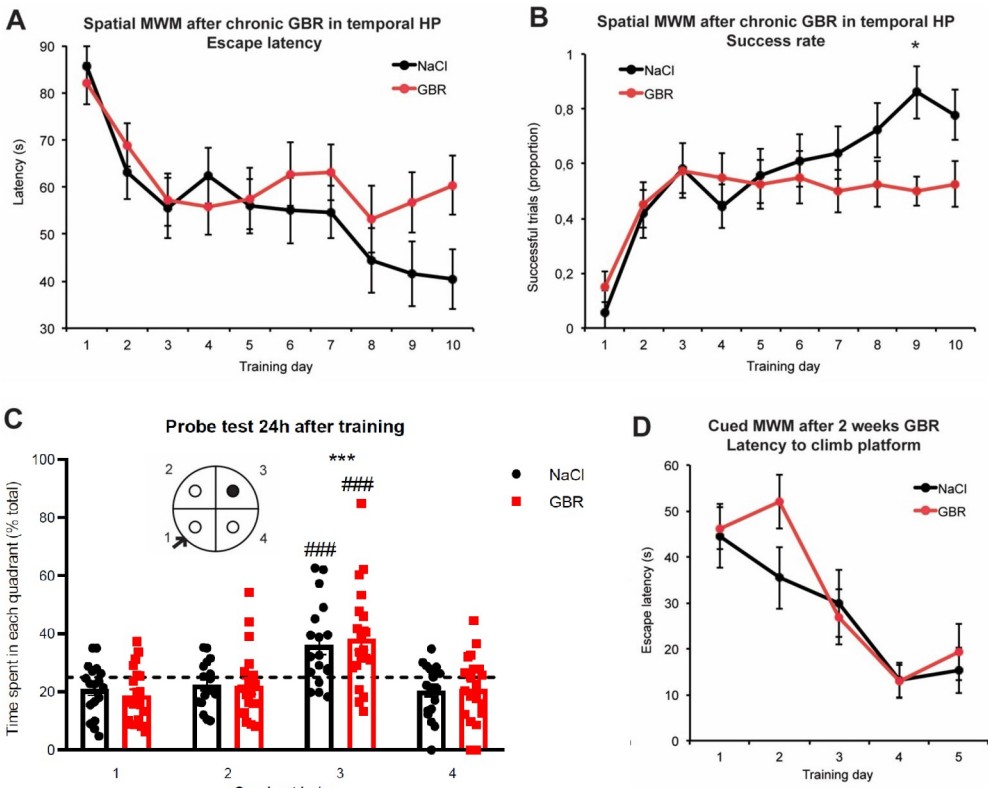

**Fig 2. Effect of GBR12935 infusion in the ventral hippocampus (CA1) on spatial and associative Morris water maze (MWM) task. A)** Mean escape latencies after GBR12935 (300nmol/L, .5μL) daily infusion in ventral CA1 (mean latency of two trials, 30 min and 90 min after GBR infusion). There was no significant difference between groups after two-way repeated measure ANOVA. **B)** Proportion of successful trials per group per day. Posthoc comparison of groups means revealed a significant difference at training day 9 (n = 18, 20; *p < .05) **C)** Probe testing of spatial memory 24h after the last training session. Both groups spent significantly more time in the target quadrant compared to the others (###p < .001 compared with 25%, ***p < .001 for Quadrant 3 compared with others; no difference between groups). **D)** Mean escape latencies in the cued/associative version of the MWM. No significant difference between groups (n = 10,10). Bars denote SEM.

OPRT (NaCl, .60±.03; GBR, .61±.03) at day 2–4 of treatment. However, GBR-treated mice displayed impaired recognition of the novel object after 16 days of infusion (NaCl, .61±.03; GBR, .52±.02, p < .05 between groups) and impaired novel object-place recognition (NaCl, .61±.02; GBR, .50±.03, p < .01 between) at day 18 of treatment. Conversely, NaCl-treated mice displayed intact recognition memory in both tests (Fig 3B and 3C). Mice from both groups displayed no preference towards either left or right object or place on the training day (S3E, S3F Fig in S1 File). The observed spatial memory deficits were not simply a consequence of emotional alterations following chronic GBR infusion into the ventral CA1, as the total distance and relative times spent in the center and periphery were not significantly different between GBR- and NaCl-treated mice when a subset of animals were tested in an open field (S3A-S3C Fig in S1 File). In keeping with this, relative thigmotaxis time was similar between GBR-infused and NaCl control mice, in spite of a significant lower rate of successful trials by GBR-treated mice on day 9 of the spatial MWM (S3D Fig in S1 File).

The short-term version of OPRT in animals treated for 3 weeks resulted in significant recognition performance in both groups. GBR- and NaCl-treated animals significantly

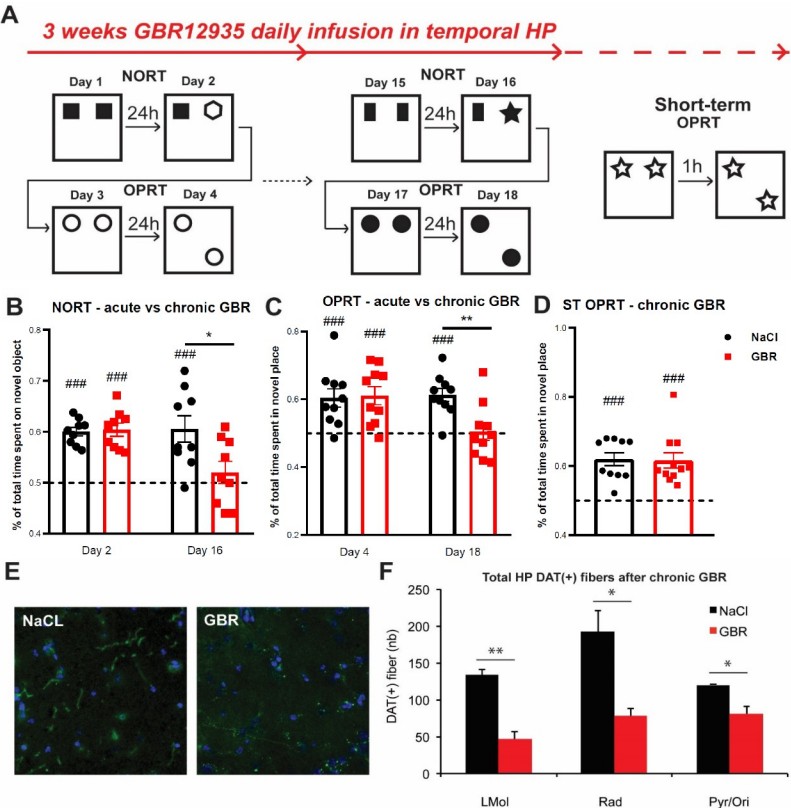

**Fig 3. Impairments in recognition memory and mesohippocampal dopaminergic fibers induced in the mouse ventral hippocampus after chronic GBR12935 infusions. A)** Design of the NORT and OPRT performed on cannulated mice between day 1 and day 18 of infusions. A short-term version (1h between habituation and testing) of the OPRT was performed on mice treated for 3 weeks **B)** NORT scores of mice after 2 or 16 days of GBR (300nmol/L, .5µL) infusion. Visuospatial retention of older object was significantly impaired in GBR-treated mice compared with NaCl-treated mice (n = 9, 9; ###p < .01 compared with .50; *p < .05 between groups) at day 16 but not at day 2 (n = 9, 9; ###p < .01 compared with .50; p = .94 between groups). **C)** OPRT scores after 4 or 18 days of GBR12935 infusion. Visuospatial retention of older place was preserved in NaCl- and GBR-treated groups at day 4 of treatment (n = 10, 10; ###p < .01 compared with .50; p = .87 between groups) but retention was impaired in GBR-treated animals at day 18 (n = 10, 10; ###p < .01 compared with .50; **p < .01 between groups). **D)** Short-term OPRT scores after 3 weeks of chronic GBR12935 infusion. Both groups equally displayed significant visuospatial retention of the older place (n = 10, 11; ###p < .01 compared with .50; p = .89 between groups). **E)** Examples of immunohistolabelling of DAT+ fibers (20X) in NaCl- and GBR-treated mice hippocampal slices (green). Co-staining was performed with Hoescht (blue) **F)** Quantification of total DAT+ fibers in hippocampi of NaCl- and GBR-treated mice (n = 3, 3). There was significantly fewer DAT+ fibers in all layers of the hippocampus of GBR-treated mice (LMol: **p < .01; Rad: *p < .05; Pyr/Ori: *p < .05 between groups). *LMol = lacunosum moleculare, Rad = stratum radiatum, Pyr/Ori = Stratum pyramidale + Stratum oriens*. Bars denote SEM.

explored the novel place more (NaCl, .60±.08; GBR, .60±.01). This observation suggests that GBR infusions in the ventral hippocampus selectively impaired long-term, but not short-term, memory of objects' spatial configuration (Fig 3D). After the last behavioral test, a subset of GBR- and NaCl-treated animals were sacrificed, and brains collected. Immuno-histolabeling of DAT+ fibers in the hippocampus (Fig 3E) revealed that chronic GBR infusion decreased drastically the number of DAT+ fibers (up to 3 fold) in all layers ((LMol) NaCl, 134±7, GBR, 47±10; (Rad) NaCl, 192±29, GBR, 78±11; (Pyr/Ori) NaCl, 120±1, GBR, 81±10) (Fig 3F).

## D2-like receptor antagonist sulpiride rescued novel-object recognition, not spatial recognition, in chronically GBR infused mice

We hypothesized that the effect of chronic excess of extracellular DA in the ventral hippocampus was essentially driven by presynaptic D2Rs. Therefore, local blockade of D2Rs achieved with typical antipsychotic sulpiride might alleviate some, or all, of the deficits observed. New C57BL/6J mice were divided in four groups and treated either by: (1) systemic NaCl (.9%, 100mL/kg) followed by intrahippocampal NaCl (.05μL); (2) systemic NaCl followed by intrahippocampal infusion of GBR12935 (300nmol/L, .5μL); (3) intrahippocampal infusion of sulpiride (15μmol/L, .5μL) followed by intraHP GBR12935 (300nmol/L, .5μL); and (4) systemic sulpiride (50mg/kg) followed by intraHP GBR12935 (300nmol/L, .5μL) (Fig 4A). Intrahippocampal sulpiride infusion served as a mechanistic means of investigating local effects of DA levels on CA1 D2Rs and systemic sulpiride administration as a model of typical antipsychotic administration.

Mice of group (1), treated with only NaCl, performed normally in every test, displaying a significantly longer exploration time for the novel object in the NORT paradigm ((1) .61±.01) and for the novel place ((1) .62±.02) in the OPRT paradigm. In accordance with the previous experiments, mice of group (2), treated with local GBR12935, failed to display exploratory preference for the novel item in either the NORT ((2) .52±.01) or OPRT ((2) .51±.01) paradigms. Groups (3) and (4), where GBR infusions were preceded by either systemic or local sulpiride, displayed opposite performances between the NORT and the OPRT test (Fig 4B and 4C); in the NORT, both groups successfully recalled the older object at day 16 of co-treatment ((3) .63±.02; (4) .63±.04). Conversely, both groups (3) and (4) failed to recognize the novel place in the OPRT at day 18 ((3) .53±.02; (4) .50±.03).

## Acute GBR12935 impaired LTD expression at CA3-CA1 synapses through D2R signalling

The constitutive model of DAT knockout (DATKO) mice revealed that excessive DA activity would lower the threshold for long-term potentiation (LTP) induction and block long-term depression (LTD) expression at Schaffer collateral-CA1 synapses [28]. To assess the acute effect of pharmacological DAT blockade, we applied GRB12935 (30nmol/L) on coronal brain slices of adult C57BL/6J mice containing ventral CA1 prior to induction of LTP and LTD at Schaffer collateral-CA1 synapses.

High-frequency stimulation induced a significant LTP in both control (1.55±.08) and GBR-treated slices (1.38±.12) (Fig 5A). LTP levels were not statistically different. Paired-pulse low frequency stimulation induced stable depression in untreated slices (.78±.08), but GBR application blocked LTD expression (1.08 ±.06) (Fig 5B).

Acute DAT blockade reproduced the LTD impairment found in DATKO mice. To assess the effect of D1Rs or D2Rs blockade on LTD expression under GBR treatment, we applied the $D_1/D_5$ antagonist SCH23390 (1μmol/L) or $D_2/D_3$ antagonist sulpiride (10μmol/L) in co-application with GBR12935 before LTD induction. Pharmacological co-blockade of $D_1/D_5$ and DAT did not affect the LTD impairment observed with GBR12935 alone (1.08±.11) (Fig 5C). Conversely, blockade of $D_2/D_3$ and DAT enabled the induction of significant LTD (.87±.03) (Fig 5D).

## Discussion

Imbalances in dopamine activity in cortical and subcortical regions are thought to underlie the pathophysiology of schizophrenia. These deficits are poorly managed by available medications

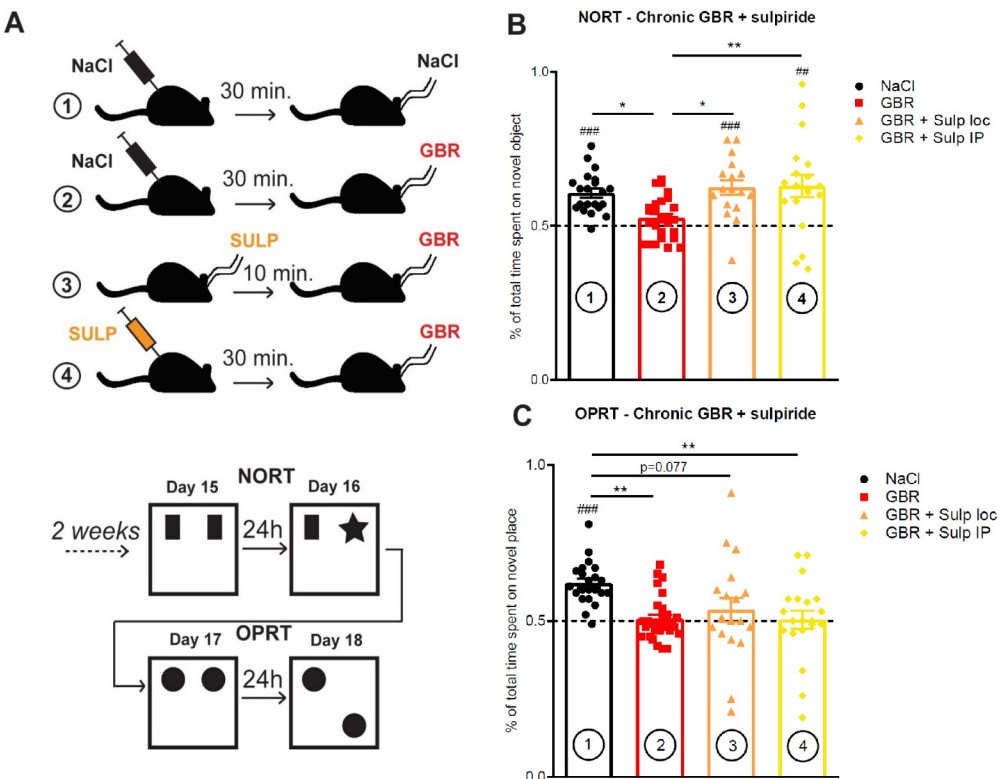

**Fig 4. Effect of D$_2$/D$_3$ antagonist sulpiride on GBR-induced recognition memory impairments. A)** Experimental design of the test. Animals were separated into 4 groups and systemically treated daily with intraperitoneal injection (IP) of saline (.9%) or sulpiride (50mg/kg). 30 min later, animals were intrahippocampally infused with either NaCl or GBR (300nmol/L, .5µL) as previously. Group (3) (sulp local) was treated intrahippocampally with sulpiride (10µmol/L, .5µL) 10 min prior to GBR12935 infusion. After two weeks of repeated daily co-treatment, performances in the NORT and the OPRT were assessed. Results are compared with one-way ANOVA followed by Tukey's HSD posthoc comparisons of the means. Mean difference with theoretical value of symmetric exploration (.50) was assessed in each group with one-sample Student t test. **B)** NORT scores after 2 weeks of co-treatment. Local GBR infusion (GBR, n = 26) impaired visuospatial retention of the older object compared to NaCl group (NaCl, n = 21) as previously described (*$p < .05$ compared with GBR; ###$p < .001$ compared with .50). Pre-treatment with either sulpiride locally (GBR+sulp local, n = 17; *$p < .05$ compared with GBR; #$p < .05$ compared with .50) or sulpiride IP (GBR+sulp IP, n = 19; **$p < .01$ compared with GBR; ##$p < .01$ compared with .50) before GBR infusion resulted in restoration of novel-object recognition. **C)** OPRT scores after 2 weeks of co-treatment. Chronic intrahippocampal GBR infusion impaired spatial recognition of the older place compared to NaCl group ((2) n = 28; ###$p < .001$ compared with .50; **$p < .01$ compared with (1)). Pre-treatment with sulpiride locally ((3) n = 18; $p = .38$ compared with .50; $p = .08$ compared with (1)) impaired spatial recognition of older place even if significant difference with NaCl group was not fully reached. Pre-treatment with systemic sulpiride ((4) n = 20; $p = .89$ compared with .50; **$p < .01$ compared with (1)) resulted in impaired spatial recognition as well. Bars denote SEM.

despite their great importance in social dysfunction that patients experience. Reduced DAT expression and human DAT polymorphism affecting its function were found to be associated with psychosis expression in bipolar disorder and schizophrenia [41]. However meta-analyses failed to find a significant association of major DAT variants with schizophrenia incidence or vulnerability [42, 43]. DAT dysfunction could still underlie specific subsets of cognitive defects seen in schizophrenia, in particular working memory deficits [44]. Few rodent models have been used to study the role of dopamine transport in the context of mental disorders. The hippocampus is a critical structure in the consolidation of episodic memory, spatial memory in rodents. It is therefore essential to study how impaired DAT function and the resulting increase in dopamine extracellular availability in the hippocampus would affect memory and

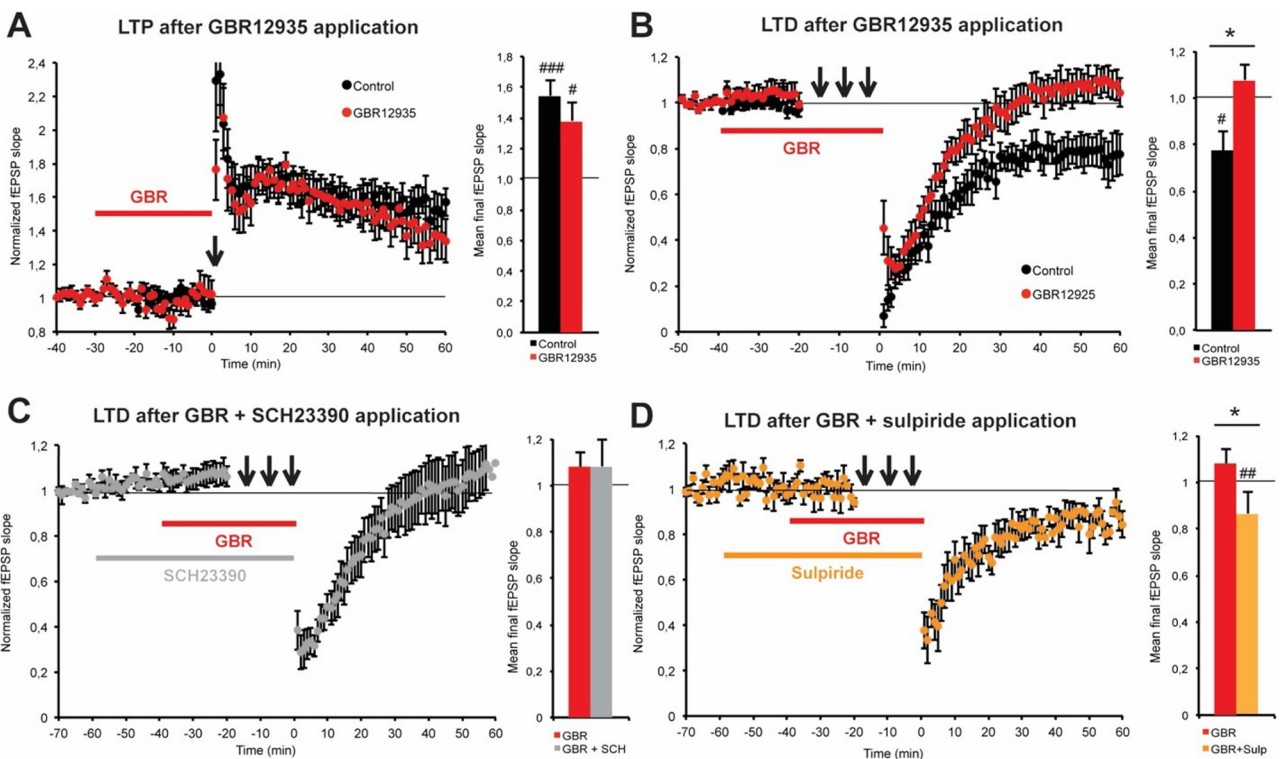

**Fig 5. In vitro acute effect of GBR12935 (30nmol/L) on LTP and LTD in ventral CA1. A)** HFS (3x100pulses/100Hz/20s) induced significant LTP in dorsal CA1 of coronal slices of naïve mice ventral hippocampus (n = 7; ###p < .001 compared with baseline). GBR application (30 min) did not significantly change LTP levels induced by tetanic stimulation (n = 7; #p < .05 compared with baseline; p = .31 between groups). **B)** Paired-pulse LFS (1200 pairs/1Hz/200ms IPI) induced significant LTD (n = 7; #p < .05 compared with baseline). GBR application (40 min) completely blocked the expression of ppLFS induced LTD at Schaffer collateral-CA1 synapses (n = 7; p = .25 compared with baseline; *p < .05 between groups). **C)** Co-application of the $D_1/D_5$ antagonist SCH23390 (1μmol/L) did not prevent GBR from blocking LTD expression (n = 6; p = .5 compared with baseline). **D)** Co-application of $D_2/D_3$ antagonist sulpiride with GBR allowed the expression of significant LTD (n = 6; ##p < .01 compared with baseline; *p < .05 between groups). Bars are SEM.

its underlying synaptic mechanisms. Unexpectedly, we clearly observed two distinct modes of DA transmission in the ventral versus dorsal hippocampus. In the ventral part there is a classical organization of DA fibers, mostly arising from ipsilateral VTA that contains the DAT for extrasynaptic clearance. In the dorsal section, not only did we not detect any DAT-labelled fibers [22] but we also found that the DA neurons innervating the dorsal hippocampus (all originating from the VTA) did not express DAT. It therefore appears that two distinct pools of DA neurons in the VTA project to the hippocampus, providing a classical wired connection in the ventral section, whereas in the dorsal section, the absence of DA reuptake will likely provide a volumic transmission [45].

The daily blockade of DAT in ventral CA1 across training days induced a mild impairment during the learning phase of the MWM. Latency to escape the platform in the GBR12935 group successfully decreased in the early phase of learning but performance reached a plateau after 5 days. Still, after 11 days of treatment, the significant exploration of the escape quadrant by both control and treated mice during probe test showed that spatial memory was properly acquired. Deficits in habituation have been extensively reported in genetic and neurodevelopmental rodent models of schizophrenia [46–50]. As this basic form of behavioral plasticity relies on novelty detection [51] we focused on a more profound investigation of this aspect of the behavior.

Long-term object recognition tasks in rodent are emotionally neutral tests and rely on intact hippocampal function [52–55]. The dopaminergic mesohippocampal pathway is strongly implicated in novelty detection [10, 11]. Both D1Rs and D2Rs in the hippocampus have been shown to modulate synaptic plasticity occurring during novelty exploration and the consolidation of novel spatial representations in mice [12–14]. We submitted ventral-CA1 GBR-treated mice to two classical object-recognition paradigms that rely on spontaneous novelty detection, the NORT and the OPRT that are recommended tasks to screen for memory impairments in animal models of schizophrenia [3]. To avoid habituation differences between groups, we reduced the tests to their core elements: a single habituation session with identical objects and a testing session 24 hours later. Chronic DAT blockade induced a total deficit in NORT and OPRT after 2 weeks of GBR infusion in ventral CA1. The converse absence of deficit observed after the first infusions suggested that a gradual development of a maladaptive process was involved, rather than a direct, fast emerging dysfunction. Importantly, even after 3 weeks of daily GBR12935 infusion, recognition in the short-term version (1 hour of delay between habituation and testing) of the OPRT was preserved. It is consistent with previous reports that support a specific role for DA in the rodent prefrontal cortex (PFC) in the modulation of working memory [4, 56]. The implication of the PFC cannot be ruled out in long-term recognition memory as several studies found a dependency of long-term NORT performances on monoamine transmission and ERK signaling in the PFC [57, 58]. Conversely, firing of CA1 neurons was observed in response to novelty in a short-term version of the NORT in rats [59].

In a parallel set of experiments, we observed no deficit either in spatial, reversal or recognition memory following chronic infusion of GBR12935 in the dorsal hippocampus. The observed dichotomy between dorsal and ventral effects of GBR12935 are likely a direct consequence of DAT expression being restricted to the ventral part of the mouse hippocampus; DAT-expressing fibers (DAT$^+$) innervate only the dorsal *stratum radiatum* and ventral *lacunosum moleculare* of ventral CA1 [22]. Finally, because the GBR concentrations we used were kept below the affinity threshold for other monoamine transporters expressed in dorsal CA1, the effects in ventral CA1 are most likely specific to the blockade of DAT-mediated reuptake.

A growing number of studies in recent years have suggested the existence of a functional continuum, related to spatial processing and navigation, along the dorsoventral hippocampal axis [60]. Loureiro and colleagues demonstrated that participation of the ventral hippocampus is substantial for the expression of a spatial memory representation [61]. In a pioneering study, Fredes et al., 2021 validated such functional continuum from the dorsal to the ventral hippocampus, demonstrating that mossy interneurons in the ventral dentate gyrus detect environmental novelty and that this information is conveyed to dorsal dentate granule cells and is necessary for novelty-induced contextual memory formation [62]. Interestingly, we previously showed that mossy cells of the dentate gyrus constitute the main D2 receptors expressing cells in the ventral hippocampus [22]. Considering a positive correlation between the activity of ventral mossy interneurons on one hand and environmental novelty and contextual memory acquisition on the other hand [62], it is conceivable that suppression of ventral mossy cells due to over-activation of their D2 receptors by dopamine excess would lead to impairment in spatial memory formation in ventrally infused mice. In support of this hypothesis, antagonism of D2 receptors by the antipsychotic sulpiride successfully rescued the spatial memory deficit induced by chronic GBR infusion.

Our observation that chronic dopamine reuptake inhibition in the ventral hippocampus induces memory defects is consistent with schizophrenia symptoms. The apparent absence of spatial memory impairments in the MWM would require a closer investigation. The time-course of GBR-induced deficits in the mesohippocampal pathway might be too slow to induce a strong impairment after only 11 days of treatment. Another hypothesis is that the

incremental learning aspect that the MWM paradigm entails could allow the mouse memory to bypass specific alterations induced in the ventral hippocampus owing to the activity of dorsal hippocampus and other structures critically involved in this task, such as the dorsomedial striatum.

Structural plasticity changes are critical for the expression of adaptive behaviors. Rodent exposure to psychostimulants provides evidence that blocking dopamine reuptake in dopaminoceptive regions induces structural changes that could impair behavioral responses and synaptic plasticity [63, 64]. D2R antagonists such as antipsychotics have also been shown to strongly regulate structural plasticity at pre- and postsynaptic dopamine sites in the basal ganglia [64–66]. We previously described how DAT-expressing fibers sprout following antipsychotic exposure in the mouse ventral hippocampus, an effect replicated by genetic ablation of presynaptic D2R [22]. Here we saw that chronic exposure to GBR12935 conversely induced a strong decrease of DAT$^+$ fibers in the mouse hippocampus, in accordance with psychostimulant action [67].

The physiological consequences of these important alterations in DAT$^+$ terminals' arbor remain to be determined. Local infusion of GBR12935 in vivo has been shown to increase extracellular dopamine levels up to 400% within tens of minutes following infusion in the nucleus Accumbens, and dopamine levels remained high for another hour after the end of infusion [36]. Continuous infusion of GBR12909 (same properties as GBR12935) in the rat hippocampus has been found to lead to a 200% rise in dopamine concentration [68]. Therefore, the huge reduction of DAT$^+$ fibers observed after chronic DAT blockade suggests that a hypodopaminergic phenotype was taking place as an adaptive response, probably due to overactivation of the presynaptic D2Rs.

As we described previously, D2-like receptors blockade by sulpiride was detrimental for learning and memory in the mouse hippocampus. Moreover, D2 knockout mice exhibit a deficit in NORT [22]. But in contrast to GBR exposure, presynaptic D2Rs blockade favours the sprouting of DA terminal in normal mice [22, 64, 66, 69]. While a significant reduction in DAT function has not been demonstrated in individuals with schizophrenia, a disruption in the physical interaction between D2R and DAT has been shown in post-mortem striatum samples of individuals with schizophrenia [70]. We hypothesized that local D2Rs blockade would preserve normal memory in mice by counteracting GBR-mediated structural and physiological deficits on DA transmission.

We co-treated the mice with either intrahippocampal or systemic sulpiride to compare a region-specific and a therapeutic-like approach. Interestingly, both local and systemic sulpiride treatments preserved novelty detection in the NORT but failed to restore spatial-based recognition memory in the OPRT. This dichotomy was surprising, as we would expect sulpiride to either succeed or fail to normalize alterations in both types of recognition deficits induced by GBR. Nevertheless, the NORT and the OPRT rely on overlapping but distinct networks. Several studies argue that novel-object recognition is not dependent on the hippocampal function *per se*, but rather, its spatial component is dependent on the hippocampus solely when the context of the objects displays a certain complexity [55, 71]. Conversely, object-place recognition depends on the spatial component of memory; according to theories of functional dichotomy along the septoventral axis, ORPT may engage dorsal CA1 networks and D5R expressed at pyramidal neurons [72–74]. Antagonising presynaptic D2R might only partially restore dopamine levels in the proximal area of DAT$^+$ fibers. This could explain why spatial recognition performance was not restored as well as novelty performance in response to antipsychotic treatment. If such hypothesis is true, OPRT scores could be more sensitive to dopamine postsynaptic D1Rs modulation in dorsal CA1.

Hippocampal glutamatergic transmission, particularly N-methyl-D-aspartate (NMDA) receptor activation, is critically involved in the consolidation of an early memory as well as in a

long-term one [75]. Perturbation of LTP/LTD at hippocampal synapses alters certain aspects of memory and novelty detection. Reciprocally, novelty exposure favors the expression of synaptic plasticity, especially LTD, in the hippocampus [71]. LTP and LTD are known to be processes highly dependent on an optimal tonic dopamine concentration. We hypothesized that a homeostatic regulation of LTD expression occurs in the mouse ventral hippocampus through D2Rs and DAT-mediated reuptake activities. In vitro, DAT blockade by GBR12935 has been shown to provoke a pathological increase of LTP magnitude in the rat CA1 region [76], an effect mediated by D3Rs through induction of gamma-aminobutyric acid-A (GABA-A) receptor endocytosis [77]. As GBR12935 chronic infusions strongly impaired recognition memory, we suspected that blocking the DAT had also an effect on long-term synaptic plasticity in mice CA1.

In a first set of experiments, we assessed the acute effect of GBR12935 on LTP and LTD at Schaffer collateral-CA1 synapses. In presence of a GABA-A receptor blocker, GBR application did not modify LTP levels. However, dopamine reuptake blockade abolished LTD expression. The discrepancy in results between Swant et al. [76] experiments and ours relies on experimental design and species specificities; we could not detect mRNA expression for D3Rs in the mouse hippocampus [22] while they did. Moreover, GBR12935 decreased basal transmission at higher doses (~μmol/L) under our conditions. We therefore selected a GBR concentration that would not affect basal transmission to avoid confounding effects. In line with our previous work, we found that LTD modulation by dopamine was dependent on D2R activity in ventral CA1. $D_2/D_3$ receptor blockade by sulpiride before GBR application rescued significant LTD expression. The mechanism by which LTD is regulated by D2Rs activity remains unclear. We have no evidence so far of direct postsynaptic regulation of pyramidal cells excitability by D2Rs in CA1. Excitability of the Schaffer collateral is also modulated by the activity of CA3 neurons, which are wired to mossy fibers arising from the dentate gyrus. D2Rs are highly expressed in mossy glutamatergic interneurons of the Hilus and we already speculated on a role for altered mossy cell activation in the observed effects of GBR12935. In our opinion, the presynaptic control over levels of dopamine remains the key to LTD regulation in the ventral CA1, probably by impacting postsynaptic D5R activity on pyramidal cells.

DAT has been suggested to be expressed by astrocytes [78] and microglia [79, 80] in vitro, although it is not established if glia in the hippocampus also express DAT. In our hands, DAT staining was limited to neuronal fibers in the ventral hippocampus and any glial staining fell below our detection threshold. Therefore, while we believe our observations mainly originates from a neuronal component in the ventral hippocampus, we cannot exclude a possibility of glial DAT contribution, to a certain extent, to the effects of GBR on electrophysiology and behavior.

Chronic blockade of the DAT in the mouse ventral/temporal hippocampus induced pathological alterations in synaptic plasticity, habituation, and long-term recognition memory consistent with cognitive symptoms of schizophrenia. Though hippocampal dopaminergic concentration is low and only few DAT$^+$ fibers innervate the ventral subsection, it reinforced our previous observation that the mesohippocampal pathway undergoes important structural plasticity adaptations, independent of early development, when dopamine levels are altered. Interestingly, the novel-object recognition deficit that chronic DAT blockade induced, was reversed by typical antipsychotic administration at both local and systemic levels, whereas antipsychotics are known to damage long-term memory in normodopaminergic mice. However, antipsychotics failed to improve spatial recognition. At the synaptic level, acute DAT blockade altered LTD, a process linked with novelty binding into memory in the hippocampus. Similarly, the deficit was counteracted by a typical antipsychotic, whereas it induced the opposite effect in healthy mice. Further investigations are required to understand the nature of

the memory deficit induced by dopamine reuptake inhibition. Contrary to the idea that anti-psychotics have little beneficial effect on cognition, this study sheds some mechanistic light on the potential utility of antipsychotics in alleviating cognitive symptoms of mental health disorders provided that the physiopathology of the disorder involves hyperdopaminergic states in the hippocampus.

## Supporting information

**S1 File.**
(PDF)

**S1 Data.**
(XLSX)

## Author Contributions

**Conceptualization:** Jill Rocchetti, Gohar Fakhfouri, Bruno Giros.

**Data curation:** Jill Rocchetti, Caroline Fasano, Gregory Dal-Bo, Elisa Guma.

**Formal analysis:** Jill Rocchetti, Caroline Fasano, Elisa Guma, Tak-Pan Wong, Gohar Fakhfouri.

**Funding acquisition:** Salah El Mestikawy, Bruno Giros.

**Investigation:** Caroline Fasano, Gregory Dal-Bo, Elisa Guma.

**Methodology:** Jill Rocchetti, Gregory Dal-Bo, Tak-Pan Wong, Bruno Giros.

**Project administration:** Bruno Giros.

**Resources:** Bruno Giros.

**Supervision:** Jill Rocchetti, Bruno Giros.

**Validation:** Tak-Pan Wong, Bruno Giros.

**Writing – original draft:** Jill Rocchetti, Bruno Giros.

**Writing – review & editing:** Gohar Fakhfouri.

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
