## [Decision Letter · Decision Letter 0]

2 Mar 2023

PONE-D-23-03664Persistent Extrasynaptic Hyperdopaminergia in the mouse Hippocampus Induces Plasticity and Recognition Memory Deficits Reversed by Antipsychotics.PLOS ONE

Dear Dr. Giros,

Thank you for submitting your manuscript to PLOS ONE. After careful consideration by 2 Reviewers and an Academic Editor, all of the critiques of both Reviewers must be addressed in detail in a revision to determine publication status. If you are prepared to undertake the work required, I would be pleased to reconsider my decision, but revision of the original submission without directly addressing the critiques of the Reviewers does not guarantee acceptance for publication in PLOS ONE. If the authors do not feel that the queries can be addressed, please consider submitting to another publication medium. A revised submission will be sent out for re-review. The authors are urged to have the manuscript given a hard copyedit for syntax and grammar as this is requisite for publication consideration.

**Comments to the Author**

1. Is the manuscript technically sound, and do the data support the conclusions?

Reviewer #1: Yes

Reviewer #2: Partly

2. Has the statistical analysis been performed appropriately and rigorously? 

Reviewer #1: Yes

Reviewer #2: No

3. Have the authors made all data underlying the findings in their manuscript fully available?

Reviewer #1: No

Reviewer #2: Yes

4. Is the manuscript presented in an intelligible fashion and written in standard English?

Reviewer #1: Yes

Reviewer #2: Yes

5. Review Comments to the Author

Reviewer #1: This is a beautiful, novel and elegant study that I loved reading, providing several unexpected observations that are tempting food for thought. This is probably the reason for several of the queries listed below:

1- Spatial reference memory has been mostly associated with information processing in the dorsal rather in the ventral aspects of the hippocampus. Instead, the ventral part of the hippocampus is mostly involved in the emotional state and emotional-related memories. Thus, there seems to be a need to discuss the apparently paradoxical effects on spatial reference memory upon selective manipulations of the ventral hippocampus. Moreover, it seems relevant to enquire if fear memory, namely extinction of fear memory, and anxiety-/depressive-like behaviors are not modified with these dopamine-related manipulations in the ventral hippocampus. With the data already collected, some information may be obtained, namely by quantifying the relative time in the center and periphery of the open field and some thigmotaxic behavior in the Morris water maze. Indeed, this clarification of eventual GBR-induced emotional alterations seems important since alterations of mood-like behavior could explain differences in the spatial-memory tests.

2-The title explicitly claims that the study explores a ‘Persistent Extrasynaptic Hyperdopaminergia’. Please provide the evidence for this blunt conclusion that this reviewer could not reach after reading the manuscript.

3-The mentioning of effects of extrasynaptic dopamine prompts questioning if DAT is present in astrocytes and/or microglia in the hippocampus and if dopaminergic effects on these glial cells can be excluded as a possible mechanism of GBR-induced electrophysiological and behavior effects.

4-The title also claims that the modifications caused by GBR treatment were ‘Reversed by Antipsychotics’. This is a rather group of drugs and only a D2 receptor-preferring antagonist was tested out of the several drugs in this class. This may need to be reviewed.

5-p.4 ‘GBR12935 has a high specificity for DAT blockade’. Specificity means that a drug only has one target irrespective of the concentration. Thus, specificity is not high or low, instead it exits or does not exist. Unfortunately, there is no single drug that is specific, they are all at best selective. Correct.

6-p.5: At least define the hippocampal synapses where the recordings were done as well as the recording medium in the core of the manuscript.

7-p.5 ‘GraphPad (Prism, which version?)’: I am afraid only the authors will be able to answer this question.

8-p.10 ‘Time spent exploring the novel item (object or place)’: I may have missed it, but I could not find the description of the control values during the training period of the treated and untreated mice: did they spend a similar total time exploring the two objects during the training phase?

9-p.12 ‘Figure 5. In vitro acute effect of GBR12935 (30nmol/L) on LTP and LTD in ventral CA1’: I would strongly advise the authors to include representative fEPSP recordings to convince the readers (and this reviewer) of the quality of the recordings, especially in the presence of bicuculline, which tends to result in heavily contaminated fEPSPs, even without knowing the constitution of the recording solution, which I would consider the minimal detail that should be provided in the core manuscript.

10-p.13 ‘memory, GBR-treated animals displayed a hyper-exploratory phenotype, possibly due to a deficit in long-term habituation’: I could not find the data sustaining this statement. It seems mandatory to request seeing these data since alterations of locomotion can be sufficient to explain differences in the memory-related behavior tests.

11-p.13 ‘memory, GBR-treated animals displayed a hyper-exploratory phenotype, possibly due to a deficit in long-term habituation’: This also raises the question of defining the searching strategy of mice without or with treatment with GBR (see Garthe and Kempermann, 2013, Front. Neurosci. 7, 63).

12-p.14 ‘Hippocampal lesions generally spare short-term recognition memory performance (50)’: This contention is at least debatable since several acute interference with hippocampal function result in profound impact on short term memory tests (60-90 min in OD or OR).

Reviewer #2: Rocchetti et al. described in this manuscript their study on the effect of persistent extrasynaptic hyperdopaminergia in the mouse hippocampus, which induces plasticity and recognition memory deficits on behavior using behavior tests and electrophysiology. They concluded that antagonizing D2Rs was beneficial for cognitive functions in the context of hippocampal hyperdopaminergia.

Although the study appears complete by its data, several major concerns include deficient experimental design and misused statistical methods, missing critical information on technical details, invalidated behavioral data, and failure to put their findings in the context of relevant non-therapeutic-oriented studies in the literature.

Major concerns:

It is necessary to provide evidence for how well the repeated administration model of GBR 12935, a dopamine transporter (DAT) blocker in the hippocampus, simulates psychiatric disorders in clinical settings. Behavioral changes can occur when dopamine function is enhanced, and results indicating indirect cognitive impairment may be obtained depending on the behavioral tests. These results naturally depend on the extracellular dopamine levels and are thought to be antagonized by antipsychotic drugs. For example, in some patients with schizophrenia, brain dopamine function is enhanced, but compensatory dopamine receptor function is thought to be decreased if dopamine levels are always high. In this animal model, it should be shown what functional changes occur due to repeated administration of GBR. For example, how long after discontinuing the infusion of the DAT inhibitor for 2-3 weeks did extracellular dopamine levels increase, and how much did dopamine neuron reactivity change before the next infusion? Additionally, it should be confirmed that there were no histological impairments.

Since it has yet to be known whether behavioral experiment data are close to the normal distribution, there may be problems presenting results with mean values and standard errors, Student t-tests that assume a normal distribution. At the very least, each data point should be shown as a scattered dot blot on the graph.

6. PLOS authors have the option to publish the peer review history of their article (what does this mean?). If published, this will include your full peer review and any attached files.

**Do you want your identity to be public for this peer review?** For information about this choice, including consent withdrawal, please see our Privacy Policy.

Reviewer #1: **Yes: **Rodrigo A. Cunha

Reviewer #2: No

We look forward to receiving your revised manuscript.

Kind regards,

Stephen D. Ginsberg, Ph.D.

Section Editor

PLOS ONE
---

## [Decision Letter · Decision Letter 1]

26 Jul 2023

Persistent Extrasynaptic Hyperdopaminergia in the Mouse Hippocampus Induces Plasticity and Recognition Memory Deficits Reversed by the Atypical Antipsychotic Sulpiride.

PONE-D-23-03664R1

Dear Dr. Giros,

We’re pleased to inform you that your manuscript has been judged scientifically suitable for publication and will be formally accepted for publication once it meets all outstanding technical requirements.

Kind regards,

Stephen D. Ginsberg, Ph.D.

Section Editor

PLOS ONE

**Comments to the Author**

1. If the authors have adequately addressed your comments raised in a previous round of review and you feel that this manuscript is now acceptable for publication, you may indicate that here to bypass the “Comments to the Author” section, enter your conflict of interest statement in the “Confidential to Editor” section, and submit your "Accept" recommendation.

Reviewer #1: All comments have been addressed

Reviewer #2: All comments have been addressed

2. Is the manuscript technically sound, and do the data support the conclusions?

Reviewer #1: Yes

Reviewer #2: Yes

3. Has the statistical analysis been performed appropriately and rigorously? 

Reviewer #1: Yes

Reviewer #2: Yes

4. Have the authors made all data underlying the findings in their manuscript fully available?

Reviewer #1: No

Reviewer #2: Yes

5. Is the manuscript presented in an intelligible fashion and written in standard English?

Reviewer #1: Yes

Reviewer #2: Yes

6. Review Comments to the Author

Reviewer #1: The authors have addressed all questions raised and provided a better balanced manuscript.

I confess that I would be uncomfortable to submit a manuscript without all the data available to be presented on request.

Reviewer #2: (No Response)

7. PLOS authors have the option to publish the peer review history of their article (what does this mean?). If published, this will include your full peer review and any attached files.

Reviewer #1: No

Reviewer #2: **Yes: **Masayuki Hiramatsu

---

## [Editor Report · Acceptance letter]

16 Aug 2023

PONE-D-23-03664R1 

Persistent Extrasynaptic Hyperdopaminergia in the Mouse Hippocampus Induces Plasticity and Recognition Memory Deficits Reversed by the Atypical Antipsychotic Sulpiride. 

Dear Dr. Giros:

I'm pleased to inform you that your manuscript has been deemed suitable for publication in PLOS ONE. Congratulations! Your manuscript is now with our production department. 

Kind regards, 

on behalf of

Dr. Stephen D. Ginsberg 

Section Editor

PLOS ONE